# De novo annotation reveals transcriptomic complexity across the hexaploid wheat pan-genome

Benjamen White [1,49], Thomas Lux [2,49], Rachel Rusholme-Pilcher [1,49], Angéla Juhász [3,49], Gemy Kaithakottil [1], Susan Duncan [1,4], James Simmonds [4], Hannah Rees[1], Jonathan Wright [1], Joshua Colmer[1], Sabrina Ward[1], Ryan Joynson[1,5], Benedict Coombes [1], Naomi Irish[1], Suzanne Henderson[1], Tom Barker[1], Helen Chapman[1], Leah Catchpole[1], Karim Gharbi [1], Utpal Bose[3,6], Moeko Okada [7,8,9], Hirokazu Handa [10], Shuhei Nasuda[11], Kentaro K. Shimizu [7,8], Heidrun Gundlach [2], Daniel Lang [2,12], Guy Naamati[13], Erik J. Legg[14], Arvind K. Bharti[14], Michelle L. Colgrave [3,6], Wilfried Haerty [1], Cristobal Uauy [4], David Swarbreck [1], Philippa Borrill [4], Jesse A. Poland [15], Simon G. Krattinger [15], Nils Stein [16,17], Klaus F. X. Mayer [2,18], Curtis Pozniak [19], 10+ Wheat Genome Project*, Manuel Spannagl[2,20] ✉ & Anthony Hall [1,21] ✉

Wheat is the most widely cultivated crop in the world, with over 215 million hectares grown annually. The 10+ Wheat Genomes Project recently sequenced and assembled to chromosome-level the genomes of nine wheat cultivars, uncovering genetic diversity and selection within the pan-genome of wheat. Here, we provide a wheat pan-transcriptome with de novo annotation and differential expression analysis for these wheat cultivars across multiple tissues. Using the de novo annotations we identify cultivar-specific genes and define the core and dispensable genomes. Expression analysis across cultivars and tissues reveals conservation in expression between a large core set of homeologous genes, in addition to widespread changes in subgenome homeolog expression bias between cultivars and cultivar-specific expression profiles. We utilise both the newly constructed gene-based wheat pan-genome and pan-transcriptome, demonstrating variation in the prolamin superfamily and immune-reactive proteins across cultivars.

Wheat (*Triticum aestivum*) is the most widely grown crop and is cultivated in 12 mega-environments across the world[1], with 777.7 metric tonnes harvested globally in 2021/22 (www.fao.org). Pressures of climate change, political instability, a move to more sustainable farming and a reduction in agricultural land are putting increasing demand on international wheat harvests[2]. Efforts to overcome these pressures can be accelerated by understanding the genetic diversity of global wheat cultivars and their pan-transcriptional variation.

Wheat has a large (15 Gb) allohexaploid (BBAADD) genome, derived from a series of relatively recent hybridisation events[3]. Its size, evolutionary history, and high repeat content, despite hindering genome assembly, make wheat an interesting model for the

A full list of affiliations appears at the end of the paper. *A list of authors and their affiliations appears at the end of the paper.
✉e-mail: manuel.spannagl@helmholtz-muenchen.de; anthony.hall@earlham.ac.uk

evolution of large polyploid genomes. Step changes in technology have enabled the chromosome-level assembly of nine high-quality wheat genomes by a global consortium. These genomes revealed evidence of widespread structural rearrangements, introgression from wild relatives and the impacts of parallel international breeding programmes[4,5]. To date, these genomes have been annotated only by projecting Chinese Spring gene models across the new assemblies. The generation of de novo annotations for these genomes provides a key insight into gene gain and loss, reveals novel gene models across wheat cultivars and facilitates comparative gene expression analysis between cultivars.

Previous analyses of the wheat transcriptional landscape described tissue-specific changes in gene expression in two cultivars, using a common Chinese Spring reference genome[6]. Polyploidy leads to complex effects on gene expression resulting from structural variation, gene duplication, deletion and neofunctionalization, ultimately increasing variation in gene expression and the plasticity of the species. The allotetraploid pan-transcriptomes of *Brassica napus*[7] and cotton[8] identified asymmetry in gene expression between subgenomes and differential gene expression between introgressed regions.

Here, we generate de novo gene annotations, incorporating long reads for the nine assembled wheat cultivars, providing a valuable resource for wheat researchers and breeders. The construction of a reference-agnostic, gene-based wheat pan-genome yields evidence of widespread gene duplication and deletion, revealing the impact of international breeding programmes on genome architecture. We define the hexaploid wheat core and dispensable transcriptome and our analysis of gene expression and gene networks across different tissues and between cultivars reveals conservation and divergence in expression balance across homoeologous subgenomes. We exemplify the value of these analyses through in-depth investigation of the pan-genome variability of prolamin gene content and expression; a key trait for quality and health aspects in wheat.

## Results

### De novo gene annotations of cultivars define the core and accessory gene sets

To precisely assess the gene content and differences in gene expression, copy number and the presence/absence of genes between the wheat cultivars, we generated a de novo gene annotation for each of the nine cultivars. A detailed description of origin, biological characteristics and diversity coverage of these cultivars was provided previously[4]. We used an established automated annotation pipeline which built evidence-based gene model predictions using a comprehensive transcriptomic dataset. This dataset was made up of Iso-Seq data from roots and shoots (390–700 K reads per sample), and RNA-seq data (150 bp paired-end read, 56–85 M pairs of reads per sample) obtained for each cultivar from five distinct tissue types and whole aerial organs sampled at dawn and dusk (Fig. 1A, see 'Methods' for a full description and Supplementary Fig. 1 for details of quality control). In addition to the transcriptomic dataset, the gene annotation pipeline also used protein homology and ab initio prediction. Finally, a gene consolidation procedure (Supplementary Fig. 2A) was developed to identify and correct for missed gene models in each specific cultivar. This step ensures the best possible comparability between the wheat genomes and gene repertoire[9].

The number of high-confidence (HC; definition provided in 'Methods') gene models identified ranges from 140,178 for CDC Landmark to 145,065 for Norin 61 (Fig. 1B). Low-confidence genes, primarily representing gene fragments, pseudogenes and gene models with only weak support, are in the range of 315,390 (Mace) to 405,664 (SY Mattis). With a maximal difference of 3.5%, the number of high-confidence genes appears to be similar across cultivars, whereas most of the differences in gene number observed can be attributed to the

low-confidence gene set. For around 70% of the HC genes, we obtained evidence for transcription in at least one condition.

We benchmarked the quality of the de novo gene predictions against BUSCO v5.1.2 with the poales_odb10 lineage dataset, representing 4896 Poales near-universal single-copy orthologs. On average, we found more than 99.8% of the BUSCO genes represented as at least one complete copy and 86% by three complete copies (Fig. 1B). This is an improvement in complete BUSCO genes over the gene projections from Chinese Spring used in the first 10+ wheat genomes study[4] and can be explained by the de novo gene annotation strategy applied here, which included comparable RNA-seq and Iso-Seq datasets and ab initio prediction, as well as the final consolidation step. Completeness and consistency of the gene sets were measured using OMArk with an average of 97.1% of high-confidence genes corresponding to gene families in the Pooideae clade and only 4.5% missing, likely classified as low-confidence (Supplementary Fig. 2C). The de novo annotations are available in Ensembl Plants release 52.

### The wheat pan-genome identifies gene duplications and introgressions

Genes and gene families exclusive or amplified in a specific cultivar are of major interest in a pan-genomic context[10]. While genes present in all compared cultivars are referred to as the core genome, cloud and shell genes are found only in one (cloud) or shared in a subset of cultivars (shell). In our study we follow terminology for the definition of pan-genome and pan-transcriptome established for other crops[11,12], although use of nomenclature is not always consistent in other plant pan-genome studies. The improved gene annotation enabled the construction of a fully reference-agnostic, gene-based pan-genome for bread wheat. It is noteworthy that our pan-genome differs in many ways from those constructed for natural populations[13], as we are investigating a hexaploid species of approximately 8000 years old that has experienced population bottlenecks, and includes cultivated material originating mainly from breeding efforts rather than natural selection and/or drift. Consequently, genes and genomic regions identified in the core, cloud and shell portions of our pan-genome mostly represent the result of targeted selection and could become important targets for wheat improvement and breeding. GENESPACE[14] was used to derive syntenic relationships between all chromosomes and subgenomes, allowing in-detail investigation of macro- and microsynteny (Fig. 1C) and gene copy number variations. While previously identified rearrangements, such as the chromosome 5B/7B translocation in SY Mattis and ArinaLrFor, were confirmed, additional frequent small-scale structural variations can now be examined in the context of their gene content. We found a 16 Mb inversion, split into three segments of around 5 Mb each, on chromosome 3D between Canadian cultivars CDC Stanley and CDC Landmark which coincides with the locations of QTLs related to biomass and grain weight[15].

We identified groups of orthologous genes (referred to as orthogroups) among the wheat high-confidence gene models of all cultivars. A total of 55,478 orthogroups contained 99.8% of all genes, with 112 orthogroups identified as cultivar-specific and 2784 genes not clustered in any orthogroup - defining the cloud genome. Cloud and shell genes have previously been found to be associated with disease resistance[16], adaptation to new environments[17], or important agricultural traits[18]. Within the shell genome, our analysis identified orthogroups that are shared only between specific cultivars. Examples include CDC Stanley and CDC Landmark from Canada, Mace and LongReach Lancer from Australia or ArinaLrFor, SY Mattis and Julius from Europe (highlighted in yellow in Fig. 2A) which all share exclusive sets of genes. These observed patterns likely reflect the complex breeding history of the selected cultivars which represent wheat lines from different regions, growth habits and breeding programmes. Inspection of the chromosomal location of these gene groups identified multiple clusters (Fig. 2B and Supplementary Fig. 3) that are likely

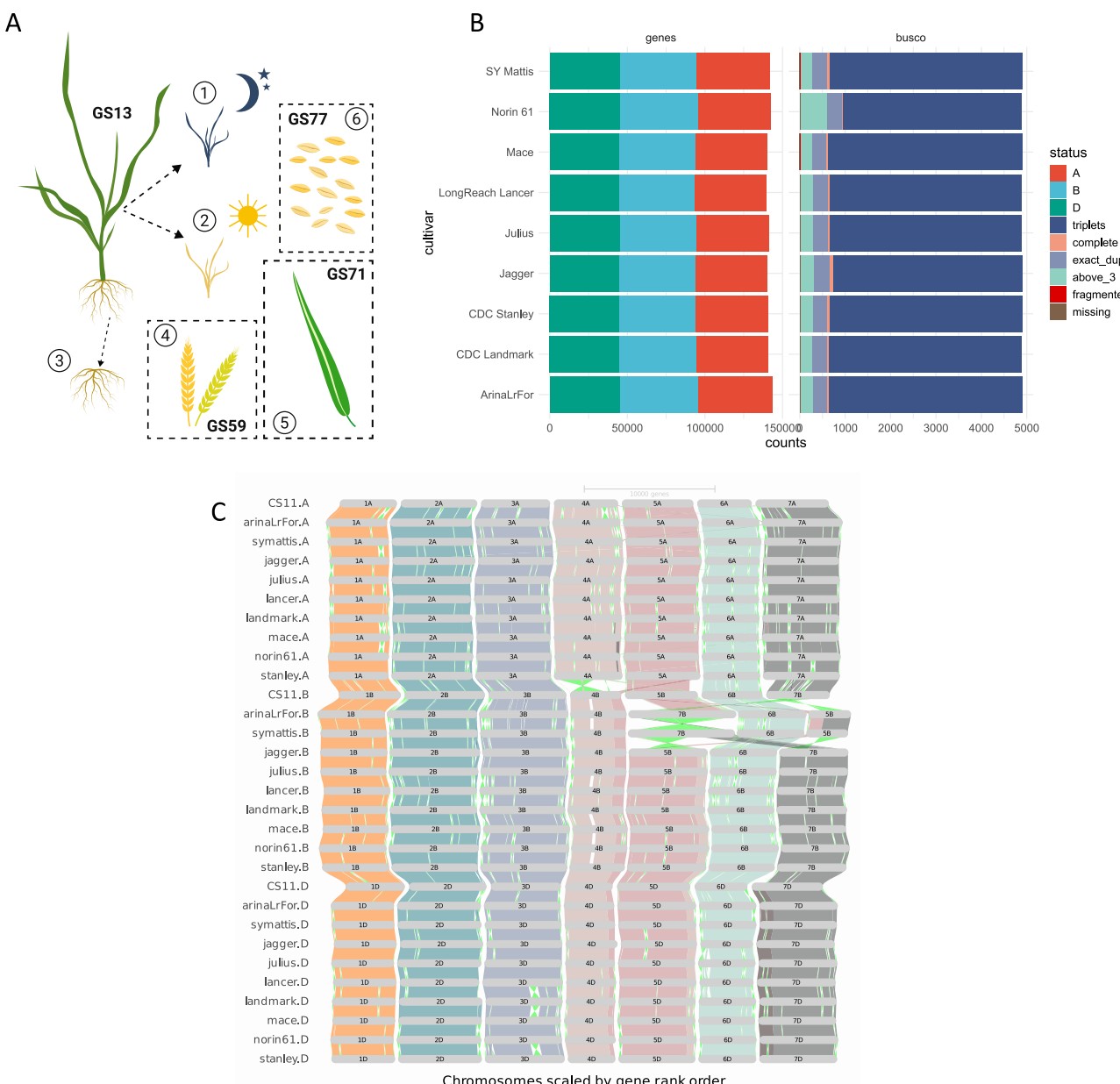

**Fig. 1 | Study design, de novo gene annotations and orthologous framework.** **A** Overview of transcriptome data generated for this study of wheat cultivars. 1 and 2: whole aerial organs sampled at dawn and dusk, 3: root, 4: complete spike at heading (GS59), 5: flag leaf 7 days post anthesis (GS71), 6: whole grains 15 days post anthesis (GS77). **B** De novo gene prediction results for each cultivar (left side, 'genes', separated for A, B and D subgenome) as well as summary of the BUSCO completeness assessment of gene models (right side, 'BUSCO'). BUSCO genes found in two copies/duplicates are referred to as 'exact_dupl' and BUSCO genes found in more than three copies as 'above_3'. **C** GENESPACE construction and visualisation of orthologous genes within the wheat cultivars, using de novo predicted gene models. Source data are provided as a Source Data file.

associated with crosses to distinct material or hybridisations with wild or domesticated relatives; events common in wheat[19].

Proportions of core (genes present in all cultivars), shell (genes present in 2–8 cultivars) and cloud (genes found in only one cultivar and unclustered genes) genes were found to be similar across cultivars (Fig. 2C). On average, 62.52% of genes were classified as core, 36.61% as shell and 0.86% as cloud (Supplementary Fig. 2B). These findings are consistent with proportions of conserved and variable genes reported by the first wheat pan-genome study[20] and a recent pan-genome constructed from mainly Chinese wheat cultivars[21]. Amongst the core gene set, we find biological functions associated with basic metabolic, catabolic and DNA repair/replication processes enriched (Supplementary Data 1), while stress response and regulation of gene expression were overrepresented in the shell genes (Supplementary

Data 2). In the set of cloud genes, functions related to chromatin organisation and reproductive processes were found to be enriched (Supplementary Data 3). These findings concur with gene functions predicted for the core-, shell- and cloud- gene sets in other crop plants[20,22]. Expression patterns of core, shell and cloud genes revealed pronounced differences globally, but not between the subgenomes. As observed in other pan-genomes[23], core genes tend to be more highly expressed in all subgenomes and tissues, as compared to both shell and cloud genes (Fig. 2D).

In order to further analyse the characteristics of cultivar-specific genes we extracted 592 genes that are unique to the Japanese cultivar Norin 61 based on our new de novo annotation (Supplementary Data 4). The genomic positions of those genes are significantly enriched ($p = 1.30E\text{-}27$, Fisher's exact test,

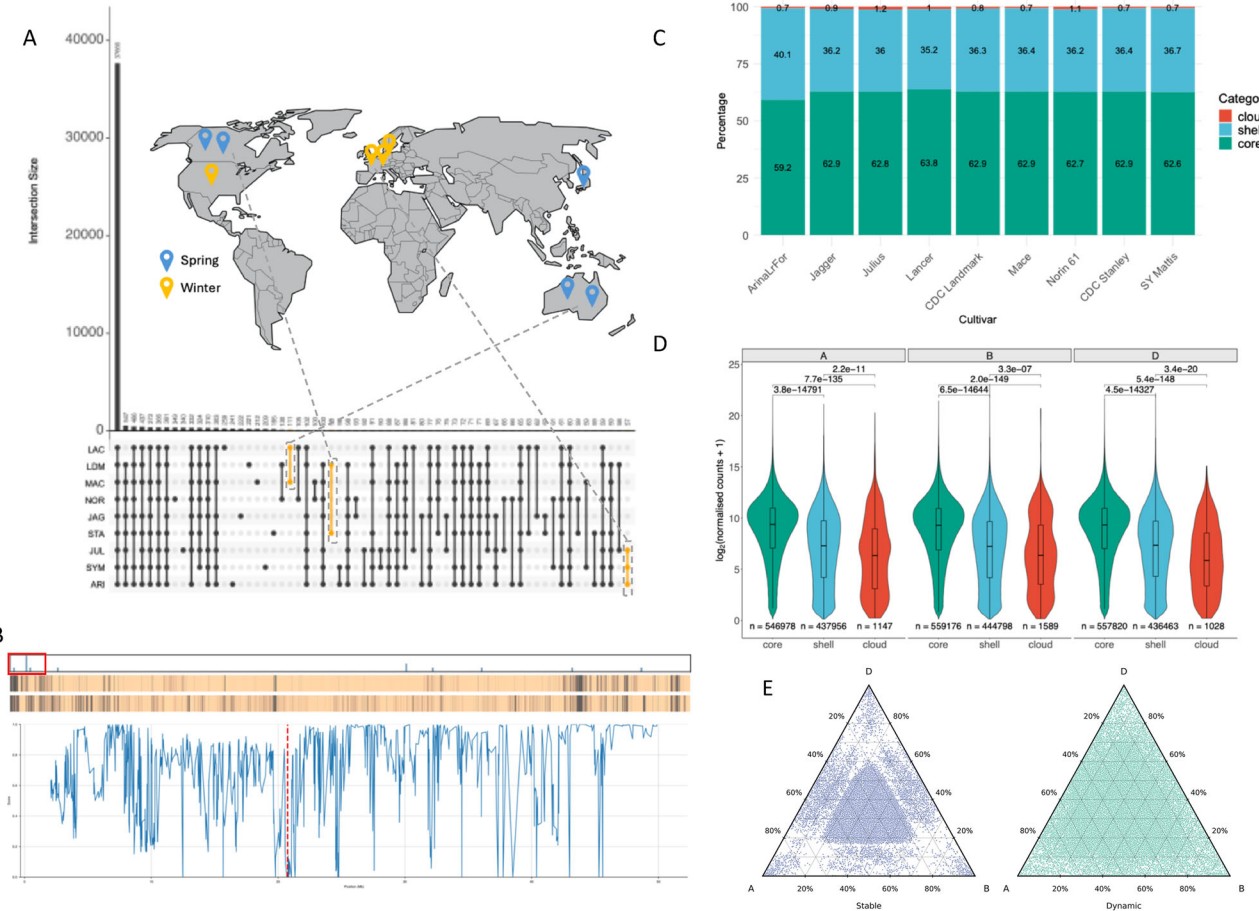

**Fig. 2 | The wheat core-, shell- and cloud- genome and homoeologous expression patterns. A** UpSet plot showing intersects of orthogroup conservation between cultivars and the relation to their breeding programmes and sowing season. Locations are at the country/state level as cultivars are representative of national breeding programmes. **B** A representation of CDC Stanley chromosome 3B showing the positions of Canadian-specific genes (top bar), heatmaps showing coverage scores between genes in CDC Stanley and CDC Landmark (middle bar) and coverage scores between CDC Stanley and Norin 61 (bottom bar). Coverage scores are calculated using kmers from each CDC Stanley gene to search the genome of the other cultivar and range from 0 to 1 with values closer to 1 indicating greater similarity. Regions of greater difference are represented in the heatmaps as darker bands. The plot shows a detailed view of the 0–50 Mb region of chromosome 3B (indicated by a red box). The mean of the coverage score between CDC Stanley genes in this region and genes in the non-Canadian lines is plotted. A cluster of four Canadian-specific genes (marked by a red dashed line) lies in a region which is noticeably different between CDC Stanley and the non-Canadian lines potentially representing an introgression. **C** Percentage of genes belonging to

the core-, shell- and cloud- orthologous groups across cultivars. **D** Violin plots of core, shell and cloud $\log_2$ average gene expression across all combined cultivars and tissues, for each subgenome. Internal box plots display the median (centre line), with boxes representing the 25th to 75th percentiles (interquartile range) and whiskers extending to 1.5× the interquartile range. Outliers are not displayed. Pairwise comparisons between categories (core vs shell vs cloud) were performed using two-sided Dunn's test for multiple comparisons following a Kruskal–Wallis test. Bonferroni correction was applied to adjust $p$-values for multiple testing. Exact $p$-values are shown above each comparison. Higher mean expression was observed in core genes across all subgenomes. **E** Ternary plots, of stable (left) and dynamic (right) 30-let (definition in main text) expression, where there is an homeolog present on each subgenome, of all tissues in all cultivars, combined, showing more overall balanced expression in stable 30-lets and unbalanced expression in dynamic 30-lets. Source data are either provided in an online repository (https://doi.org/10.5281/zenodo.16964999)[78] or as a Source Data file (Fig. 2C).

Supplementary Data 5) in 176 Norin 61 unique genomic regions (Supplementary Data 6). Furthermore, they were similarly enriched in 45 chromosomal segments that were previously shown to be specific to Norin 61 based on the unique pattern of transposable elements (TEs) and include putative alien introgressions[4,24,25] ($p$ = 0.000264, Fisher's exact test; Supplementary Data 5 and 7). We next determined the expression patterns of those genes and found that 202 of the 592 Norin 61-specific genes were expressed at least in one of the examined tissues in Norin 61 (Supplementary Data 4), and more than half of them (107) showed tissue-specific expression (tau > 0.8), excluding transposable elements (Supplementary Data 8). Among the genes with tissue-specific expression patterns, functional descriptions related to defence response are enriched (Gene Ontology analysis; Supplementary Data 9). Within the

genomic regions with Norin 61 specific TE patterns, we find genes related to disease resistance and plant immunity, such as *NON-EXPRESSER OF PR GENES 1* and *RGA2*. These results demonstrate that the cultivar-specific chromosomal segments not only harbour unique coding and TE sequences consistent with introgressions but also encode genes with specific expression patterns, suggesting that these segments are promising targets for finding genes responsible for unique traits such as disease resistance.

Duplication of genes has been identified as a major driver of gene function evolution and adaptation in plants[17]. In wheat, a large number of tandem duplications was previously found both in the Chinese Spring (IWGSC v1.1)[26] reference genome and the 10+ wheat assemblies[4]. Our full de novo gene annotation of the 10+ wheat genomes, in combination with the extensive gene expression data

presented in this study, allowed for an in-depth assessment of gene duplication dynamics across cultivars in hexaploid wheat.

We identified on average 7,011 tandem arrays (HC genes only) in each cultivar, with the lowest in CDC Landmark (6889) and ArinaLrFor as the highest (7196). We further classified these tandems as to whether they are true tandems (exactly two copies) or part of a tandem array (more than two genes). In the first category, there are on average 2918 and in the latter 4093.

In addition, we tested whether there is a bias in expression towards one member of the array. We found that for the true tandem arrays 2384 (88%) one of the two members was biased in its expression with respect to the other member, whereas for 534 (12%) arrays both copies were expressed at similar levels (Supplementary Fig. 2D). For arrays of tandems 904 (19%) showed a balanced expression, whereas 3188 (81%) were imbalanced.

Amongst all tandemly duplicated genes in wheat, biological functions associated with phosphorylation, response to stimulus and stress and reproductive processes were enriched (Supplementary Data 10). Interestingly, biological processes for responses for stress, stimuli and toxic substances were enriched within the groups of balanced true tandems. Protein modification processes, regulation of biosynthetic processes and regulation of gene expression were enriched in the unbalanced true tandems.

We also investigated the conservation of tandem arrays across all cultivars. MCScanX[27] was used to construct co-linear chains of tandems shared by two or more cultivars. For the resulting 2950 tandem chains (at least two, and up to nine cultivars in the chain) we investigated the conservation of their expression profiles. We found 1567 tandem chains with conserved/constant expression patterns across the tandems in the chain (that is either all balanced or imbalanced) and 1383 tandem chains with variable expression profiles. These results highlight the impact of tandemly duplicated genes as a potential key driver of evolution and adaptation. Besides functional redundancy of homoeologous genes in hexaploid wheat, tandem genes and their expression (bias) are therefore an important target for breeding applications.

## Conservation of global expression in the wheat pan-transcriptome

To investigate changes in global gene expression across cultivars, biological replicates from whole aerial organs at dusk and dawn, and from flag leaf, root, spike and grain, were aligned to the corresponding de novo gene models to generate normalised gene expression counts. We observed from principal component analysis of the normalised counts that most of the variance is represented by the first principal component, representing the different developmental stages and also similar grouping of expression overall (Supplementary Fig. 1). We then used these normalised counts from the nine cultivars together with complete de novo annotations for the core, shell and cloud group genes, to explore differences in expression between tissues across all cultivars. The patterns of expression observed in each individual orthologous class were consistent across tissues, and between sub-genomes, with core genes showing an overall higher mean expression than either shell or cloud (Fig. 2D, Supplementary Fig. 4). Cloud average gene expression was observed to be significantly lower than both core and shell, irrespective of tissue type or sub-genome biases ($p$-adj $< 0.05$). Overall, there is a globally conserved pattern of expression.

The tissue-specific gene index (tau) was employed to assess the degree of gene expression specificity to flag leaf, root, spike or grain tissues across all cultivars (Supplementary Fig. 5A). We observed the least number of tau genes in flag leaf (1005–3202 specific genes), that were significantly less ($t$-test; $p < 0.001$) overall compared to either root (4736–8974 specific genes), spike (5453–9323 specific genes) or grain (3955–12,157 specific genes), that showed no significant

difference between each other. However, the number of specific genes showed the least cultivar variability for flag leaf tissues, compared to the wide range in the number of grain-specific genes observed between cultivars. This could be the result of contrasting transcriptomic complexity between flag leaf and grain tissues, representing different developmental stages of maturity and metabolic activity. In polyploid crops, agricultural traits are often modulated by an interaction of homoeologous copies of genes[6]. In wheat, previous studies have focused on tissue-specific expression across homoeologous triads, identifying sets of triads that are either balanced or unbalanced in their sub-genome expression. Here, we compared variation in triad expression across cultivars using all 13,521 identified sets of 30-let genes (30-lets are defined as genes present as a triad in all nine cultivars and Chinese Spring) with a homoeolog present on each sub-genome of the de novo annotated cultivars. Using previously reported cut-off values[6], we observed similar sub-genome expression in these 30-lets, in each of the cultivars, to that reported previously in Chinese Spring, with 102 also being classed as not expressed (Supplementary Fig. 5B)[6]. However, when comparing the bias of sub-genome expression, we observed 8,028 (59.37%) of these 30-lets to have a conserved, 'stable', balanced expression between the three homoeologous copies across all cultivars. Whereby 'stable' expression relates to a conserved sub-genome expression bias between cultivars, as opposed to a 'dynamic' expression where a change in sub-genome expression bias can be observed in one or more cultivars (5052 (37.36%) of 30-lets).

As well as conservation of the balanced state, we also see conservation in dominance or suppression within triad groups, with 276 showing stable suppressed expression and 63 stable dominant expression. Stably expressed 30-lets showed GO term enrichment for essential biological processes associated with photosynthesis, translation, DNA replication, exocytosis, glycolytic process and cell redox homoeostasis (Supplementary Fig. 6A). Whilst the 5052 37.36% 'dynamically' expressed 30-lets that showed a change in the bias of sub-genome expression in at least one cultivar were found to be significantly enriched for transmembrane transport, response to stress, response to oxidative stress, defence response and photosynthesis. These dynamic 30-lets were observed to be less fixed to a specific sub-genome expression pattern compared to stably expressed 30-lets, showing a further Euclidean distance from a, b, c or centroid points (Fig. 2E). Across these dynamically expressed 30-lets, 4467 showed balanced expression in at least one cultivar, with B sub-genome suppression being the next most represented balance of expression occurring in 1972 of the dynamic 30-let sets (Supplementary Fig. 6B). Overall, more suppression of expression was seen than dominance. The Kruskal-Wallis test, applied to assess differences in the mean values of the dynamic 30-let bias across the cultivars, revealed no significant differences when examining the total percentage of each expression bias ($p > 0.05$). This suggests that the bias of dynamic expression, whilst different for individual 30-lets, has been proportionally conserved across these cultivars.

## Conserved patterns of co-expression

To explore how regulatory networks are conserved across tissues and cultivars of the pan-transcriptome, we selected four cultivars (ArinaLrFor, Jagger, Julius and Norin 61) that encompassed the range of ancestral groups represented by the wheat pan-genome modern cultivars (AG 1, 2 and 5)[28] and constructed co-expression networks. For each cultivar, we used alignments to the corresponding de novo high-confidence gene models, retaining genes expressed at greater than five normalised counts in a minimum of two samples. The resulting genes (ArinaLrFor: 102,748; Jagger: 95,162; Julius: 98,435 and Norin 61: 97,734) were used to build four cultivar-specific networks. Between 14 (Norin 61) and 18 (Julius) modules were identified for each network, accounting for between 38.4% (Jagger) and 46.1% (ArinaLrFor) of the genes in each cultivar-specific dataset (Supplementary Data 11 and 12).

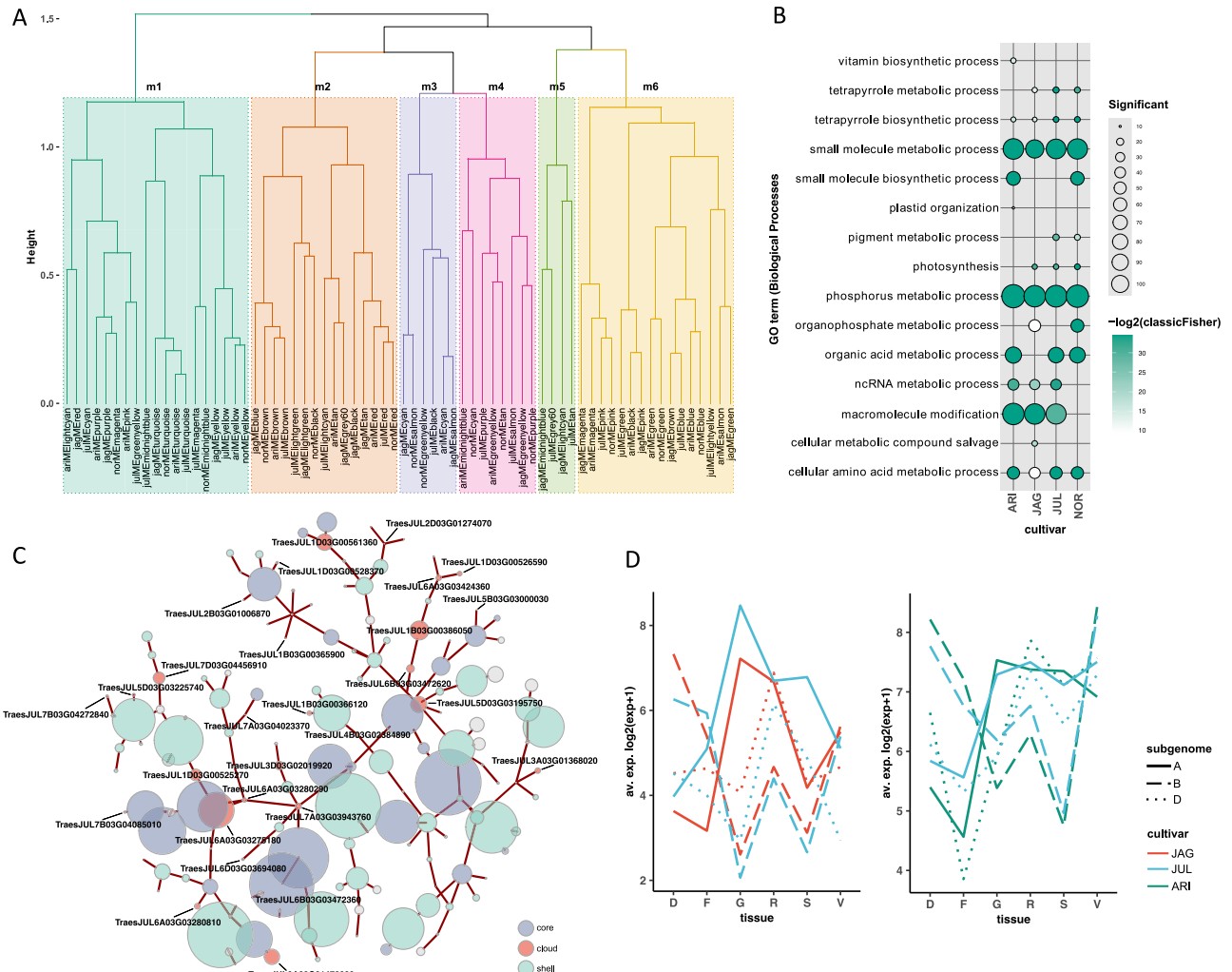

**Fig. 3 | Components of the cultivar-specific networks with functional annotation and cultivar specific differences. A** Hierarchical clustering of 68 module eigengenes from four cultivar networks identifying six metamodules (m1–m6). Each branch corresponds to a separate cultivar network module (ARI: ArinaLrFor, JAG: Jagger, JUL: Julius, NOR: Norin 61). **B** GO terms of biological processes associated with genes in conserved metamodule one. Only terms with p-adj < 0.05 and >10 significant genes are shown. Bubble colour indicates the −log2 p-value significance from Fisher's exact test and size indicates the frequency of the GO term in the underlying EBI Gene Ontology Annotation database (larger bubbles indicate more general terms). **C** Network fragment from Julius module significantly enriched for cloud genes. Labelled nodes refer to cloud genes annotated as histones. The top five highly connected genes for each cloud gene are coloured according to core or shell genome membership. Node size is scaled to the log2 average expression +1 of each gene across tissues and edge width reflects the weight of the connection between nodes. **D** Expression of two divergent 30-let triads (L: HOG0029794, R: HOG0020263) with similarly divergent subgenome patterns of expression between Jagger and Julius (HOG0029794) and ArinaLrFor and Julius (HOG0020263). Annotated as F-box transcription factor and LRR protein, respectively. Tissues D: dawn, F: flag leaf, G: grain, R: root, S: spike, V: dusk. Source data are provided as a Source Data file.

We used hierarchical clustering of 68 module eigengenes (ME) from all cultivar networks to identify six consensus metamodules (Fig. 3A) spanning the four de novo cultivar networks. Five of the six metamodules contained modules from all cultivars, demonstrating conservation of expression patterns across both tissue type and cultivar. The orthologous framework described above was used to compare the gene content of these modules. Between 19.4–40.5% of orthogroups and 60.5–72.2% of GO terms present in the three largest metamodules (m1, m2 and m6) were common to ArinaLrFor, Jagger, Julius and Norin 61 modules (Supplementary Data 13) demonstrating cross-cultivar conservation of gene content and function. GO term enrichment analysis revealed that these three conserved metamodules were involved in core processes such as photosynthesis, phosphorus metabolism, nucleosome assembly, cellular component organisation and primary metabolic processes (Fig. 3B, Supplementary Fig. 7A, Supplementary Data 14).

Two smaller metamodules, m3 and m4 had fewer GO terms in common (23.7 and 27.5%, respectively) and less than 5% of OG in

common to all four cultivars. Whilst the expression patterns of genes belonging to all cultivars were similar, GO term analysis for metamodule three (m3) revealed that Jagger, Julius and Norin 61 were significantly enriched for genes involved in organic acid biosynthesis and ArinaLrFor genes were enriched for different biological processes including phenylpropanoid metabolism (Supplementary Data 14, Supplementary Fig. 7B). Phenolic compounds are important for plant structural integrity and are implicated in the biosynthesis of key biotic and abiotic defence compounds[29]. Metamodule four (m4) had the smallest intersection of OG and GO terms between cultivars, and this was reflected in diverse GO term enrichment for each cultivar (Supplementary Fig. 7C). The final metamodule (metamodule five; m5) only contained modules from Jagger and Julius and was enriched for genes involved in intracellular transport. Jagger genes in this metamodule were most significantly enriched for molybdopterin processes whereas Julius genes were primarily enriched for genes associated with cellular localisation and positioning.

## Co-expression of cloud genes

We identified modules within our four cultivar-specific networks with a significant over-representation of previously defined cloud genes (Supplementary Data 15). Four of these modules (one per cultivar), containing more than 10 cloud genes, demonstrated highly correlated MEs (Pearson correlation coefficient >0.9, Supplementary Data 16) and could be identified within the same metamodule (m2), revealing their conservation in expression across tissue types and cultivar. The majority of cloud genes in these four modules (69–86.7%) were annotated as histones and were located in clusters on the genome. Histones have an essential role in transcriptional regulation and chromosome stability, with variants of paramount importance in the regulation of plant growth and development, and responses to both biotic and abiotic stresses[30]. Their co-expression in the cloud sector of all four cultivars could be indicative of related functions conferring cultivar-specific flexibility and adaptation. We were able to query our network resource to identify genes most highly connected to these histone orthologues for each cultivar. Visualising these network fragments for Julius revealed the connectivity of cloud genes to core and shell components and illustrated how the histone-annotated cloud genes were an integral part of the whole network (Fig. 3C, Supplementary Data 17 and 18).

## Cultivar-specific expression of 30-let genes

For each cultivar network we compared inter-module relationships to identify network modules with divergent or similar patterns of expression. We then used these module relationships to compare how the 30-let triads were split across modules within each of the four cultivar networks. Of the 14,864 conserved 30-lets, we were able to identify between 39.1% (Norin 61) and 42.9% (ArinaLrFor) of 30-lets present as complete triads within each network. Of these triads, most were assigned to modules defined as the same or similar, reflecting triad conservation in both gene expression across tissues and orthology (ArinaLrFor 96.8%; Jagger 96.1%; Julius 97.1%; and Norin 61 96.7%).

The remaining triads (ArinaLrFor 3.2%; Jagger 3.9%; Julius 2.9%; and Norin 61 3.3%) spanned divergent network modules (Supplementary Data 19). Divergent triads in the Jagger and Julius networks were significantly enriched for GO terms associated with phospholipid biosynthesis; compounds implicated in signalling pathways and regulation[31] and ArinaLrFor and Julius were enriched for genes involved in transcription elongation (Supplementary Data 20). Comparing these divergent triads sets across cultivars showed that these patterns of divergent subgenome expression were mostly cultivar-specific with >71% divergent triads demonstrating network specificity. The remaining 93 triads exhibited divergent subgenome expression in two or more cultivar networks (Supplementary Fig. 8) and contained genes with a range of functions, including leucine-rich repeats, DNA double-strand break repair and transcription factors, including five belonging to the F-box superfamily (Fig. 3D, Supplementary Data 21). F-box proteins play regulatory roles in protein degradation in response to cellular signals during plant development and growth, hormone responses and biotic/abiotic stress responses[32].

Our work demonstrates the strength of a comparative network approach in identifying potentially biologically conserved pathways between cultivars. Combining these networks with our orthologous pan-genome framework enabled the identification of genes conserved across cultivars and tissues in both function and expression. We were also able to uncover co-expression modules demonstrating cultivar-specific patterns of expression, indicating diversity and scope within each cultivar for adaptation and flexibility. The resources we generated also enabled us to place these cultivar-specific variations in a wider genomic context through the identification of highly connected network members.

## Uncovering variation in the prolamin superfamily and immune-reactive proteins across cultivars

Prolamins represent a large superfamily in wheat involved in stress responses, cell growth and plant development, as well as end-use quality and protein content[33]. Additionally, along with HMW-glutenins, prolamins trigger immune reactions in a subset of the population[34]. Here, we investigated the qualitative and quantitative differences in the 687 genes from the prolamin superfamily and HMW-glutenins to uncover their variations across the newly generated wheat pan-genome and pan-transcriptome data. We observed clear expression differences for individual developmental stages and between wheat cultivars for many genes from the prolamin superfamily, highlighting spatiotemporal variation in expression profile (Fig. 4A).

Comparison of potential immune reactive genes identified in the Chinese Spring reference genome (IWGSC v1.1) and across cultivars[24,33,35] highlighted the challenges of precise annotation and characterisation of the dynamically expanded prolamin gene families in cultivars[36–38]. This analysis therefore, utilises the manually curated Chinese Spring prolamin annotations[33] as the common reference for comparative gene expression and adds the transcriptomes (generated in the same way as for the de novo annotated cultivars; details in 'Methods') of five additional wheat cultivars without chromosome-scale genome assemblies and de novo gene predictions. The expression patterns of potentially immune-reactive gene products indicated differences in the major allergens and antigens (glutenins and gliadins). For example, SY Mattis and LongReach Lancer showed lower gene expression levels in alpha and gamma gliadins. Subsequent gene set enrichment analysis highlighted gamma gliadins as primarily enriched in the downregulated genes (Supplementary Fig. 9; Supplementary Data 22 and 23).

Celiac disease (CD) related epitope sequences encoded in the gliadin and glutenin genes show a significant variation in the wheat sub-genomes[33,35] and their generated immune response[34]. We analysed CD epitopes across cultivars and found varying expression levels of HLA-DQ epitope-containing genes. Notably, SY Mattis and LongReach Lancer exhibited lower expression, while Cadenza and Jagger showed higher expression. Cultivar-specific analysis reveals that ArinaLrFor and SY Mattis have lower alpha gliadin HLA-DQ epitope expressions due to differences in the expression activities of the three sub-genomes, possibly influenced by variations in cis-regulatory mechanisms and related transcription factor gene expressions (Fig. 4B, C, Supplementary Data 24–26; Supplementary Fig. 10). Although the sub-genome-specific expression patterns of gamma gliadin HLA-DQ epitopes did not show significant variation, the expression levels of alpha-gliadin genes with HLA-DQ epitopes from the A genome were lower in SY Mattis and LongReach Lancer. In addition, the highly immunogenic D genome alpha-gliadin epitope expression levels were lower in the ArinaLrFor cultivar (Supplementary Fig. 11A). Our results indicate that fine-tuned sub-genome-specific balance in the expression profiles may be associated with differences in the regulatory transcription factor profiles (Fig. 4B, C, Supplementary Data 26). Correlation analysis highlights common regulatory genes such as the PBF DOF triad (TraesCS5A02G155900, TraesCS5B02G154100, TraesCS5D02G161000) in complex with SPA-bZIP TFs regulate the expression of a range of prolamin and starch synthesis genes through the endosperm box[39,40], including the epitope-rich alpha and gamma gliadin genes, while different subsets of NAC or MYB genes impact the alpha and gamma gliadin gene expression (Supplementary Data 26). TF binding site motif enrichment analysis shows gliadin-type and epitope group-specific motif enrichments, offering potential targets for genome editing (Supplementary Data 27).

Gliadin and glutenin loci were found to be highly conserved in all cultivars, with some variation due to the presence of pseudogenes and gene duplications (Supplementary Fig. 11B). Reverse translated

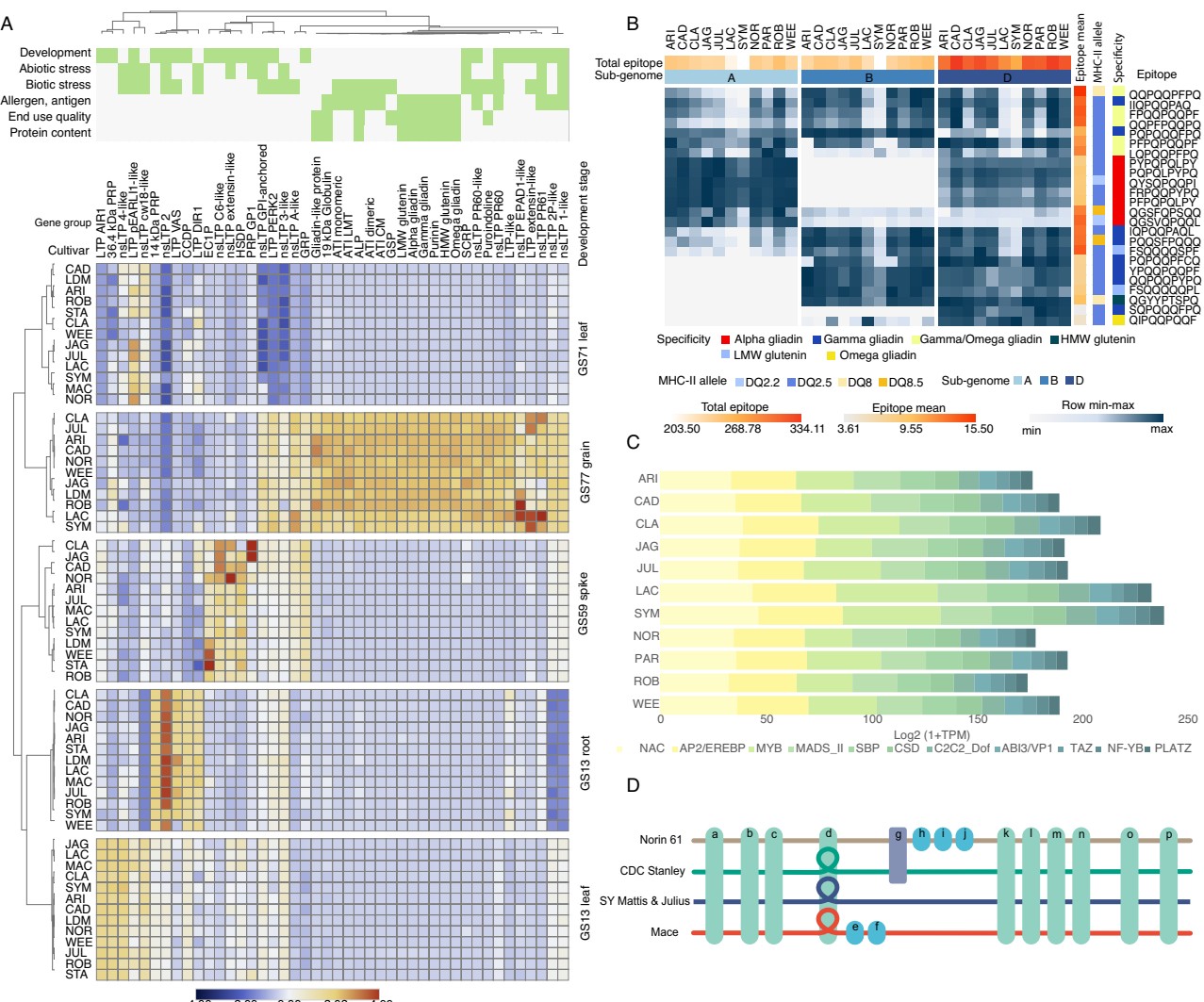

**Fig. 4 | Gene and expression variation in the prolamin family across the wheat pan-cultivars. A** Prolamin superfamily gene expression across cultivars; **B** Coeliac disease epitope expression across cultivars. Epitope expression profiles were calculated as the sum of gene expression profiles with the highlighted HLA-DQ epitopes for each subgenome. **C** Relative proportion of cumulative expression profiles of transcription factor families showing strong co-expression patterns (Pearson correlation values > 0.8) with the epitope-coding prolamin genes. Results show significant differences in the NAC, AP2/EREBP and MYB transcription factor gene expressions, major storage protein gene expression regulators. **D** Representation of the variation graph for the region of 6D containing the alpha-gliadin locus (Supplementary Fig. 11B). Horizontal coloured lines depict paths through the graph

for each cultivar; Norin 61 (6D: 26,703,647-27,222,360 bp), CDC Stanley (6D: 28,164,601-28,660,350 bp) and Mace (6D: 26,808,846-27,298,593 bp), with SY Mattis (6D: 26,645,382-27,096,594 bp) and Julius (6D: 26,983,100-27,437,565 bp) sharing a single path. Rectangular blocks (a-p) represent individual genes at corresponding locations across cultivars (green: in common to all four cultivars, blue: occurring in one cultivar and purple: occurring in two cultivars). Gene d is present as a single copy in Norin 61, and duplicated in CDC Stanley, SY Mattis, Julius and Mace. This duplication is represented as a loop in the path through the graph for these cultivars (Supplementary Fig. 12). Source data are either provided as a Source Data file or in an online repository (Fig. 4D; https://doi.org/10.5281/zenodo.16964999)[78].

consensus sequences of the known CD-specific T-cell epitopes were mapped to the genomes of all cultivars to determine the number and location of gliadin and glutenin genes containing CD-related immune reactive peptide regions (Supplementary Fig. 11B, Supplementary Data 28). The number or combination of epitopes in the loci was not significantly different between cultivars. However, the gamma gliadin and alpha gliadin gene models with a high number of epitopes were found in cultivars ArinaLrFor, Norin 61 and Mace, respectively (Supplementary Fig. 11B, Supplementary Data 28).

Although highly conserved in their locus structure on chromosome 6D, alpha gliadin genes encoding highly immunogenic proteins showed copy number variation within the wheat pan-genome. We constructed a localised pan-genome graph from five cultivars (Norin 61, CDC Stanley, SY Mattis, Julius, Mace) and extracted the subgraph of

the alpha gliadin-containing locus (Fig. 4D, Supplementary Fig. 12, Supplementary Data 29). Inspection of the subgraph helped to resolve the complex structure of the locus, with copy number variation observed as a loop in the paths of SY Mattis, Julius, CDC Stanley, Mace (2 copies of alpha gliadin genes), but not within the Norin 61 path (single alpha gliadin copy). While in total 4 to 6 epitopes were identified in the alpha-gliadins of the wheat pan-genome cultivars, 8 epitopes were detected in cultivars Mace and Norin 61 (Supplementary Fig. 11B). The gene expression profiles were found to be conserved with regard to their contribution to the total locus expression but can vary substantially for the individual genes between cultivars (Supplementary Data 30). These results indicate that gene copy number expansion primarily affected the centre of the locus and resulted in the increase of gene variants with high epitope counts. While genome-

wide construction and interpretation of pan-genome graphs remains a daunting task for complex genomes such as wheat, we found localised subgraphs, augmented by our de novo annotations, particularly helpful in resolving complex loci and uncovering structural variation, as also demonstrated in the current draft human pan-genome[41].

## Discussion

We have built de novo gene annotations for nine wheat assemblies representative of global breeding programmes[4]. Our consolidated gene annotation approach generated a robust set of core, high-confidence genes shared across cultivars. It also identified genes and gene families that are found exclusively in or amplified in cultivars derived from specific breeding programmes. It is likely that some of this variation has come through widespread introgression events[42], often associated with adaptation to biotic or abiotic stress[18]. Our annotations also identified cultivar-specific variation in tandem gene duplication. Novel gene content, gene duplication and neo-functionalisation together with gene expression patterns will have impact on researchers and breeders as they identify genes underlying traits, manipulate gene expression or incorporate and track new genetic variation.

Our analysis of global gene expression identified sets of genes with stable homoeologous expression patterns between cultivars, demonstrating tightly regulated key biological processes. We also identified homoeologous triads diverging in their expression patterns between cultivars, revealing genes enriched for processes associated with biotic and abiotic stress. Understanding the regulatory networks driving these altered patterns will provide important targets for manipulating these processes. Using network analysis, we identified widespread conservation of expression patterns across tissues and cultivars before focusing on cultivar-specific gene sets, to reveal networks of genes involved in regulation of plant growth and development and in responses to both biotic and abiotic stresses. These cultivar-specific network changes may be the result of wheat breeding programmes targeted to local environments. We also demonstrated the utility of our new resources by investigating genomic variation in the prolamin superfamily, focusing on immunogenic potential.

In conclusion, this study reveals layers of hidden diversity spanning our modern wheat cultivars. This diversity is likely to underpin the agronomic success of wheat over a wide range of global mega-environments.

## Methods

### Genome datasets used

This study builds on foundational genomes, analysis and datasets[4]. Here, we constructed full de novo gene predictions and an ortholo-gous framework (defining core/shell/cloud gene sets) for 9 wheat cultivars with chromosome-scale genome assembly sequences: Ari-naLrFor (abbreviated as 'ARI' throughout the manuscript), Jagger (JAG), Julius (JUL), Lancer (LAC), CDC Landmark (LDM), Mace (MAC), Norin 61 (NOR), CDC Stanley (STA) and SY Mattis (SYM).

The prolamin analyses utilise the manually curated Chinese Spring prolamin annotations[33] as the common reference for comparative gene expression and adds the transcriptomes (generated in the same way as for the de novo annotated cultivars) of five additional wheat cultivars without chromosome-scale genome assemblies and de novo gene predictions: Cadenza (CAD), Claire (CLA), Robigus (ROB), Weebil (WEE) and Paragon (PAR).

### Plant materials and growth conditions

The wheat cultivars were grown in a Controlled Environment Room (CER) (Conviron BDW80; Conviron, Winnipeg, Canada) set at 16 h day/8 h night photoperiod (300 µmol m$^{-2}$ s$^{-1}$, lights on at 05:00, lights off at 21:00), temperatures of 20/16 °C, respectively, and 60% relative humidity. Plants were sampled in triplicate at the 3-leaf stage

(Zadoks GS13), harvesting whole roots and whole aerial organs separately, 4 h after dawn (09:00). Whole aerial organs were also sampled 2 h after dusk (23:00). Plants for subsequent adult plant sampling were treated according to their vernalisation requirements. In the case of spring wheat cultivars (CDC Landmark, CDC Stanley, Paragon, Cadenza, Mace and LongReach Lancer), seedlings were grown as described above. At 3-leaf stage, seedlings were transferred to 1 L pots containing Petersfield Cereal Mix (Petersfield, Leicester, UK) and maintained under the same CER conditions. For winter wheat cultivars (Julius, Jagger, ArinaLrFor, Robigus, Claire and SY Mattis) and a facultative spring cultivar (Norin 61), seedlings were transferred in 40-well trays (7 days after sowing) to a vernalisation CER running at 6 °C with 8 h day/16 h night photoperiod for 61 days. After this period the plants were transferred to 1 L pots containing Petersfield Cereal Mix (Petersfield, Leicester, UK) and moved to the same CER and settings as described for the spring wheat cultivars. For both spring and winter wheat cultivars, three additional samples were harvested: complete spike at heading (GS59), flag leaf 7 days post anthesis (GS71) and whole grains 15 days post anthesis (GS77). All samples were harvested 4 h after dawn (09:00), and a single plant was used per each of the three biological replicates.

### Sample preparation and sequencing

Total RNA was extracted using Qiagen RNeasy Plant Mini Kit (cat. no. 74904) and DNAse treated using an Invitrogen TURBO DNase kit (cat. no. AM2238) according to the manufacturer's protocol. Bead purification of the RNA was conducted using the Agencourt RNAClean XP beads.system (cat. no. A63987). Final sample concentrations were verified using a Qubit 4 Fluorometer, and the integrity of the RNA was checked on the Agilent 2100 Bioanalyzer, using the RNA 6000 nano kit (Agilent, 5067-1511), running the plant total RNA assay. The directional RNA-seq libraries were constructed using the NEBNext Ultra II Directional RNA Library prep for Illumina kit (NEB, E7760L) utilising the NEBNext Poly(A) mRNA Magnetic Isolation Module (NEB, E7490L) and NEBNext Multiplex Oligos for Illumina (96 Unique Dual Index Primer Pairs) (cat. no. E6440S/L) at a concentration of 10 µM. The final libraries were equimolar pooled, a q-PCR was performed and the pool was sequenced on a Illumina NovaSeq 6000 with 150 bp paired-end reads.

The Iso-Seq libraries were constructed from 1 µg of total RNA per sample and full-length cDNA were then generated using the SMARTer PCR cDNA synthesis kit (Takara Bio Inc, 639506). The libraries were sequenced on the Sequel Instrument v1, using 1 SMRTcell v2 per library. All libraries had 600-min movies, 120 min of immobilisation time and 120 min pre-extension time.

### Data quality control and sample validation

We used a set of cultivar specific SNPs to confirm the cultivar origin of each replicate and the developmental stage of each sample was validated through a machine learning approach trained using the pooled RNA-seq samples and then run on the entire set of biological replicates. Principal component analysis of the pooled samples shows them to cluster by developmental stage as expected.

### Gene annotation

For the structural gene annotation of the chromosome-scale assembled cultivars, we combined de novo gene calling and homology-based approaches with RNAseq, Isoseq and protein datasets. The RNAseq data were mapped using STAR[43] (v2.7.8a) and further assembled into transcripts by StringTie[44] (v2.1.5, parameters -m 150 -t -f 0.3). PacBio Iso-Seq transcripts were derived from the raw reads using PacBio SMRT Link software (v5.1.0.26412rev2, pbsmrtpipe.pipelines.sa3_ds_isoseq2, default parameters). The Iso-Seq transcripts were aligned to the genome assemblies using GMAP[45] (v2018-07-04). To assist the homology-based annotation

approach, Triticeae protein sequences from publicly available datasets (UniProt, https://www.uniprot.org, 05/10/2016) were aligned against the genome sequence assemblies of all cultivars using GenomeThreader[46] (v1.7.1; arguments -startcodon -final-stopcodon -species rice -gcmincoverage 70 -prseedlength 7 -prhdist 4). All transcripts derived from RNAseq, IsoSeq and aligned protein sequences were combined using Cuffcompare[47] (v2.2.1). Stringtie (version 2.1.5, parameters --merge -m150) was employed to merge all sequences into a pool of candidate transcripts. To identify potential open reading frames and to predict protein sequences within the candidate transcript set, TransDecoder (version 5.5.0; http://transdecoder.github.io) was used.

We used Augustus[48] (v3.3.3) for the ab initio gene prediction. Guiding hints based on the RNAseq, protein, IsoSeq and TE datasets described above were used to counteract potential over-prediction[49]. Augustus was run using the wheat model.

A consolidated set of gene models was selected using mikado[50], as implemented in the Minos pipeline (https://github.com/EI-CoreBioinformatics/minos), with models scored and selected based on a combination of intrinsic qualities and support from transcriptome and protein alignments.

BLASTP[51] (ncbi-blast v2.3.0+, parameters -max_target_seqs 1 -evalue 1e-05) was used to compare potential protein sequences with a trusted set of reference proteins (Uniprot Magnoliophyta, reviewed/Swissprot, downloaded on 3 Aug 2016; https://www.uniprot.org). This approach was employed to differentiate gene candidates into complete and valid genes, non-coding transcripts, pseudogenes and transposable elements. This step was assisted by PTREP (Release 19; http://botserv2.uzh.ch/kelldata/trep-db/index.html), a database of hypothetical proteins containing deduced amino acid sequences in which internal frameshifts have been removed in many cases. We selected best hits for each predicted protein from each of the three databases used. Only hits with an e-value below 10e-10 were considered. Functional annotation of all protein sequences predicted in our pipeline was performed with the AHRD pipeline (https://github.com/groupschoof/AHRD).

We classified predicted proteins into two confidence classes: high and low confidence. Hits with subject coverage (for protein references) or query coverage (transposon database) greater than 80% were considered significant and protein sequences were classified as high-confidence based on following criteria: protein sequence was complete and had a subject and query coverage above the threshold in the UniMag database or no BLAST hit in UniMag but in UniPoa and not PTREP; a low-confidence protein sequence was incomplete and had a hit in the UniMag or UniPoa database but not in PTREP. Alternatively, it had no hit in UniMag, UniPoa, or PTREP, but the protein sequence was complete. In a second refinement step, low-confidence proteins with an AHRD-score of 3* were promoted to high-confidence.

BUSCO[52] (v5.1.2.) software was used to evaluate the completeness and accuracy of structural gene predictions with the 'poales_odb10' database containing a total of 4896 single-copy genes. OMArk[53] (v0.3.0) was also used to evaluate the consistency of the gene models against gene families in the Pooideae clade. The evidence-based part of the annotation pipeline is available at Github (https://github.com/PGSB-HMGU/plant.annot).

## Consolidation

Pairwise whole genome alignments were generated using lastz[54]. The resulting alignments were stitched together into a single whole genome alignment using TBA/multiz[55]. The MAF output was converted into HAL format using maf2hal[56].

De novo gene annotation from one cultivar was lifted over to all other cultivars using the whole genome alignment and the halLiftover tool, whereas only full-length gene models were kept. Missing gene models in one cultivar were identified using bedtools[57].

## Tandem array detection

Tandem arrays were identified using the tandem discovery model from the MCScanX package[27], with the following definitions: TrueTandems contain exactly two gene copies and TandemArrays are a sequential array of tandemly duplicated genes. Collinear tandems (chains) were detected using the detect_collinear_tandem_arrays tools provided by MCScanX and the results were filtered for TrueTandems. As described in the Alignment and Gene Expression Analysis section we used stringent mapping procedures to ensure only uniquely mapped reads were quantified, ensuring that we were able to correctly attribute reads to each respective gene copy. Expression bias was calculated using a modified method[6]. Here we used normalised read counts instead of TPM values and a cut-off of 0.8. The following categories were assigned: unbalanced for tandems with only one gene expressed and no expression data for second gene; or for tandems in which only one gene is expressed under all RNAseq conditions; balanced, where both array members are equally expressed.

## Orthogroup analysis

Two runs of OrthoFinder[58] were performed to construct the orthologous framework and both the triads and 30-lets datasets (see respective sections for definitions). The first OrthoFinder run used the full/unfiltered set of de novo predicted HC genes (subgenomes separated and treated as individual genomes) plus the Chinese Spring IWGSC RefSeq v1.1 annotation to construct 30-lets and triads. The second OrthoFinder run used the TE- and plastid-filtered set of de novo predicted HC genes to construct an orthologous framework for the pan-genome analyses. The resulting HOGs (Hierarchical Orthologous Groups) were used to determine the core-, shell- and cloud- gene sets. The longest isoforms from high-confidence genes were used as input for Orthofinder[58]. We first applied filtering criteria for TE- and plasmid-related gene descriptions. Orthofinder was run using standard parameters. We used the UpSetR in the R package (http://gehlenborglab.org/research/projects/upsetr/) to analyse and visualise how many orthogroups are shared between the cultivars or are unique to a single species. GENESPACE[14] was used to derive and visualise syntenic relationships between all chromosomes and subgenomes. Scripts for the definition of core-, shell- and cloud- gene sets were deposited at Github (https://github.com/PGSB-HMGU/BPGv2).

## Analysis of canadian-specific genes

Taking each genome in turn as a reference, kmers of length 51 were identified from genic regions using the annotation for that reference. These kmers were used to search the genomes of the other cultivars and a coverage score was computed[59] between each gene in the reference and every other genome. The coverage score (a value between 0 and 1) can be used as a proxy for sequence similarity/difference between genes in different cultivars where values closer to 0 indicate greater difference and values closer to 1 indicate similarity. Coverage scores for genes along chromosomes were plotted using the seaborn visualisation library[60] in Jupyter notebook. Coverage scores were also visualised as heatmaps with coverage scores close to 0 represented as dark bands.

## Alignment and gene expression analysis

Samples were aligned to the chromosome-scale assembled cultivars, using HISAT2 v2.0.4[61] and Stringtie v1.3.3[62] was used to extract and quantify uniquely mapped reads at gene level to the respective de novo gene models. Normalised counts were generated using DESeq2[63].

## GO term analysis

Functional enrichment of genes for biological processes was performed using the gene ontology enrichment analysis package, topGO[64] in R (v3.6.0, with the following parameters: nodeSize = 10, algorithm = 'parentchild'. Enrichment of GO terms was tested using a one-sided Fisher's exact test. *p*-values were adjusted for multiple testing using the Benjamini-Hochberg method and GO terms with adjusted *p* < 0.05 were considered significantly enriched. GO terms refer to ontology terms for biological processes unless otherwise specified and were obtained from Ensembl Plants 51, using the BioMart tool. Bubble plots were plotted using ggplot in R, adapting code from ref. 65.

## Tissue specific index

Specificity of gene expression to developmental stages was determined using the tissue specific index[66]. Where, *N* is the number of developmental stages (condition), and $x_i$ is the expression profile component for a given gene in each condition, normalised by the maximal expression value of the given gene from all conditions considered. This allowed us to classify genes as being highly specific to one condition (tau => 0.8). Assignment of tau values was performed in R using code adapted from previous work[67].

## Subgenome expression bias

Analysis of subgenome expression focused on 30-let homoeologs with a 1:1:1 relationship across all three subgenomes. Of these, 13,521 were determined to be macrosyntenic, belonging to the same subgenome in all cultivars (excluding UK cultivars which are not assembled), and 10,653 as microsyntenic, belonging to the same chromosome and subgenome in all cultivars (excluding UK cultivars). From these 66 30-lets were not taken forward in the analysis due to low expression and/or quality filtering determined by DESeq2 (R package v 4.0.3) of at least one homoeolog in each set. Relative expression of 30-lets across homoeologs and associated subgenome expression biases were calculated as previously reported, through use of our triad.expression R package (https://github.com/AHallLab/triad.expression).

## Analysis of Norin 61-specific genes and expression pattern

Norin 61 unique genomic regions were defined as regions with conserved ratios of < 0.5 among nine cultivars: Stanley, SYMattis, ArinaLrFor, Jagger, Julius, Lancer, Landmark, Mace and Norin 61. The average conservation ratio was calculated using a 2 Mbp window and 5 kbp step size. Fisher's exact test was used to determine whether Norin 61-specific genes were enriched in these Norin 61 unique genomic regions. Furthermore, unique TE regions of Norin 61 identified by the pattern of TEs were analysed similarly[24]. Tissue specificity of gene expression (tau) was calculated using the expression levels of Norin 61 using the same method described above. Genes with a tau value > 0.8 were considered as genes with tissue-specific expression. GO enrichment analysis was performed by using R Bioconductor package topGO version 2.54.0[64] using the elim algorithm and Fisher's exact test with FDR < 0.05, with 195 expressed Norin 61-specific genes excluding transposable elements. Only GO terms associated with more than ten genes were considered.

## Co-expression analysis (WGCNA)

The WGCNA R package[68] (R version 3.6.0) was used to build co-expression networks for four cultivars (ArinaLrFor, Jagger, Julius and Norin 61). These cultivars encompass the range of ancestral groups represented by the wheat pan-genome modern cultivars (AG 1, 2 and 5)[28]. The expression matrices for each of the selected cultivars contained DESeq2 normalised counts of high confidence genes derived from alignment to the respective chromosome level assemblies and corresponding de novo annotations. These matrices were filtered and genes where the sum of counts across all samples was greater than 5 in at least 2 samples were retained. We used WGCNA to construct signed networks for each cultivar using the blockwiseModules function. A soft power

threshold of 9 (ArinaLrFor, Jagger) or 10 (Julius, Norin 61) was used, together with the following parameters; minModuleSize = 30, corType = bicor, maxPOutliers = 0.05, mergeCutHeight = 0.3, minKMEtoStay = 0.4, maxBlockSize = 35,000. Eigengenes were then extracted for each module from each of the resulting four cultivar networks.

## Defining threshold for classifying inter-module relationships

To classify inter-module relationships and identify modules with divergent or similar patterns of expression we defined a threshold of module similarity. Initially we calculated the distance between each pairwise module comparison for each cultivar network, using the Pearson correlation distance. For each cultivar we used the maximum distance of each of these pairwise comparisons, for each module and calculated the medians of these maxima. Next, we investigated the proportion of 30-let triads identified as split across network modules, that would be classed as divergent using a module similarity threshold of 0-100%. From these results we selected a module similarity threshold of 85% the median of maximum distances, with distances above this classed as divergent and distances below, classed as similar.

## Identifying metamodules

We used the R package clValid[69] to determine the optimal number of clusters for the 68 ME from across all networks. The resulting Dunn index[70] and silhouette width[71] indicated that the optimum number of clusters for our ME dataset was 6. We calculated the pairwise Pearson correlation coefficients for all our 68 cultivar ME (cor()) and converted this to a dissimilarity matrix (as.dist()). We used hierarchical clustering of this dissimilarity matrix (hclust()) to define metamodules.

## Visualising highly connected cloud genes within the Julius network

We identified a network module both significantly enriched for and containing the highest number of cloud genes (29; JULbrown) and used the adjacency function of WGCNA to determine the network adjacency of each of these cloud genes within the JULbrown module. We selected the five most highly connected module genes to each cloud gene and used the R package igraph[72] to visualise the integration of these cloud genes into the JULbrown module. Using the graph adjacency function, graph adjacencies were created based on the Pearson correlation distances between genes in pairwise fashion. These directed graphs were simplified to remove multiple edges and loops, filtered to retain only those connections with an absolute Pearson correlation > 0.9. The mst function using the prim algorithm was used to create a minimum spanning tree and the resulting subgraph was visualised using the plot function with isolated nodes excluded.

## Reference allergen identification and chromosome 6D comparison

Reference allergens in the wheat pan-genome were filtered using blastn algorithm against the identified sequences in the IWGSC v1 gene annotation v1.1[33]. To identify unannotated gliadin and glutenin gene models and to compare the potential immune reactivity of the wheat cultivars, known CD-associated HLA-DQ T-cell epitopes were reverse translated, and the consensus nucleotide sequences were used for a motif search with 100% sequence identity. The mapped epitope-rich regions were used to compare the alpha-gliadin locus in chromosome 6D. Additional gene models representing complete gene models with DQ epitopes were manually annotated. The locus organisation was compared to the Chinese Spring chromosome 6D alpha gliadin locus in the IWGSC v1 reference genome assembly[33].

## Epitope expression analysis

The epitope expression values were calculated by multiplying the DESeq normalised count values of genes where the reverse-translated consensus epitope sequence was detected by the number of epitopes

in each sequence. The resulting values were then added together for each epitope type, as well as epitope types at the genome level.

## Promoter motif enrichment analysis

Gene models representing the major alpha-gliadin and gamma-gliadin gene models were selected for the comparative analysis. To investigate the impact of transcriptional regulation at subgenome levels, epitope group-specific 1000 bp promoter sequence lists were extracted from the homoeologous chromosome 1 and 6 groups of the A, B and D subgenomes using the gene models of the chromosome assembled cultivars.

Transcription factor binding site (TFBS) enrichment analysis was performed using the simple enrichment analysis (SEA) algorithm in the MEME suite[73], with the non-redundant plant-specific JASPAR 2022 motif collection for mapping. As a control promoter sequence list, 1000-bp promoter sequences were extracted from the Chinese Spring reference genome[26] high-confidence gene models, and 1% randomly sampled promoter sequences representing gene models across the 21 chromosomes were used.

TFBS enrichment results were filtered at an adjusted $p < 0.05$ using the method proposed by Benjamini and Hochberg[74] and motif hits characteristic of 100% of the analysed promoter sequences (100% true positive list), were used further analysed. Motifs showing gluten gene family type (alpha or gamma gliadin) and subgenome specificity were also highlighted (Supplementary Data 27). Motif enrichment ratios of each motif-epitope group pair were visualised in a clustered heatmap.

## Gliadin gene co-expression analysis

Gene models with known TF functions were filtered using the existing gene annotations[6,26]. Correlation values between DESeq normalised counts >1 log2 transformed DESeq counts of glutenin and gliadin gene models and the TF gene models expressed in the grain tissue were calculated in R. Hits with $r > 0.8$ and adj. $p < 0.01$ were filtered and used for further analysis (Supplementary Data 2). Wheat orthologs of representative TF genes identified through the enriched significant TFBSs were filtered and matched with the representative promoter motifs. The grain-specific co-expression network was created in Cytoscape (version 3.10.2) using a co-expression cut-off value of 0.8. The resulting network was annotated with the reference allergen-specific information for disease-relatedness and gene family. The first neighbour network was visualised in Cytoscape (Supplementary Fig. 10).

## Pan-genome graph construction of 10 Mb 6D region

We extracted a 10 Mb region (20–30 Mb) encompassing the alpha gliadin locus from the top of chromosome 6D for the cultivars Norin 61, CDC Stanley, SY Mattis, Julius and Mace. To estimate the divergence of the input sequences, we used mash-2.2[75], specifically the mash triangle command, to calculate a maximum sequence divergence of 0.039. To account for possible underestimation of sequence divergence and localised structural variants, we specified a minimum mapping identity value (-p 90) for pangraph construction using PGGB[76] together with segment size (-s 30 kb), number of mappings (-n 6), minimum length of exact matches (-k 311), target sequence length for POA (-G 13117, 13219), mean length of each sequence pad for POA (-O 0.03) and k-mer size for mapping (-K 111). Default settings were used for all other parameters.

## Extracting the alpha-gliadin locus sub-pangraph

Using ODGI toolkit[77] we extracted the subgraph of the alpha-gliadin locus from our 6D graph build. We used the odgi extract command together with coordinates of the Norin 61 gene models described in Supplementary Data 29 to extract the 520.7 kb region encompassing the locus (6D: 26,703,647-27,222,360 bp) and the corresponding paths intersecting with this region in CDC Stanley (6D: 28,164,601-28,660,350 bp), SY Mattis (6D: 26,645,382-27,096,594 bp), Julius (JUL 6D: 26,983,100-27,437,565 bp) and Mace (6D: 26,808,846–27,298,593 bp). We used odgi sort to sort the resulting subgraph and odgi procbed to adjust the coordinates of the gene models for each cultivar to fit the resulting subgraph. odgi inject allowed us to visualise the placement of these gene models across the graph and identify cultivar-specific haplotypes. We generated a graphical fragment assembly (gfa) of this subgraph[78] using odgi view.

## Reporting summary

Further information on research design is available in the Nature Portfolio Reporting Summary linked to this article.

## Data availability

The genome sequence and gene annotations of all wheat cultivars can be viewed and downloaded in Ensembl Plants (https://plants.ensembl. org/Triticum_aestivum/Info/Cultivars). This includes the de novo genes for the chromosome level cultivars generated within this study, and projected genes for all assemblies from the IWGSC RefSeq v1.1 annotation. All read data used in this study is available at the European Nucleotide Archive under accession PRJEB51827. Relevant data and results, including the gene sets for core/shell/cloud genome and wheat '30-lets', have been deposited online (https://doi.org/10.5281/zenodo. 16964999)[78]. Source data are provided with this paper.

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

## Acknowledgements

We would like to acknowledge Technical Genomics and Core Bioinformatics Groups at Earlham Institute funded by BBSRC National Capability in Genomics and Single Cell Analysis (BBS/E/T/000PR9816) and BB/CCG1720/1 for the physical HPC infrastructure and data centre delivered via the NBI Research Computing group. Part of this work was delivered via the Earlham Institute Strategic Programme Grant Decoding Biodiversity BBX011089/1 (BBS/E/ER/230002B). Delivering Sustainable Wheat BB/X011003/1 (BBS/E/ER/230003C). Norwich Research Park Biosciences Doctoral Training Partnership grant BB/M011216/1. We also would like to thank the Australian Research Council Centre of Excellence for Innovations in Peptide and Protein Science (CE200100012) and Coeliac Australia (G1005443) as well as MEXT JSPS KAKENHI 22H05179, 22H02316, 22K21352, Swiss National Science Foundation 310030_212551, Japan Science and Technology Agency JPMJCR16O3, URPP Evolution in Action and UZH Global Strategy and Partnerships Funding Scheme of the University of Zurich. Helmholtz Munich would like to acknowledge support by BMBF project number 031B0190 (de.NBI). We would also like to thank the 10+ Wheat Genome Project for their feedback and guidance. Finally we would like to thank Sasha Stanbridge (Earlham Institute) for the artwork in Fig. 1A.

## Author contributions

C.P. arranged supplying seeds. J.S. and C.U. managed growing plants and supplying materials. S.D. and H.R. prepared samples and performed extractions for sequencing. N.I., S.H., T.B., H.C., L.C. and K.G. generated and managed the iso-seq and RNA-seq data production. B.W., R.J., B.C. and J.C. performed quality control of biological replicate data. S.W. and B.W. reworked the existing package for subgenome expression bias characterisation. B.W. carried out gene expression analysis, tissue specific indexing and investigation of subgenome biases. T.L., G.K. and D.S. performed de novo annotations. T.L., D.L. and H.G. consolidated de novo annotations, generated orthogroups and investigated introgressions and copy number variation. R.R.P. performed co-expression analysis, constructed regulatory networks and gliadin pangraph, J.W. performed genome kmer comparison. A.J. and U.B. performed analysis of allergen and prolamin diversity, M.C. supervised the prolamin analysis. K.K.S., M.O., H.H. and S.N. investigated Norin 61 specific gene expression patterns. G.N. imported and managed data for EnsemblPlants. B.W., T.L., R.R.P., J.W., C.U., M.S. and A.H. contributed to preparing the manuscript. P.B., W.H., C.U., N.S., S.K., J.P., K.M. and C.P. provided intellectual input. E.L. and A.B. provided funding. The 10+ Wheat Genome Project, E.L., A.B., A.H. and M.S. conceptualised the project.

## Competing interests

The authors declare no competing interests.

## Additional information

[1]Earlham Institute, Norwich Research Park, Norwich, UK. [2]PGSB Plant Genome and Systems Biology, Helmholtz Center Munich, German Research Center for Environmental Health, Neuherberg, Germany. [3]Australian Research Council Centre of Excellence for Innovations in Peptide and Protein Science, School of Science, Edith Cowan University, Joondalup, WA, Australia. [4]John Innes Centre, Norwich Research Park, Norwich, UK. [5]Limagrain Europe, Clermont-Ferrand,

Auvergne-Rhône-Alpes, France. [6]CSIRO Agriculture and Food, St Lucia, QLD, Australia. [7]Department of Evolutionary Biology and Environmental Studies, University of Zurich, Zurich, Switzerland. [8]Kihara Institute for Biological Research, Yokohama City University, Yokohama, Japan. [9]Graduate School of Science and Technology, Niigata University, Niigata, Japan. [10]Graduate School of Life and Environmental Sciences, Kyoto Prefectural University, Kyoto, Japan. [11]Graduate School of Agriculture, Kyoto University, Kyoto, Japan. [12]Bundeswehr Institute of Microbiology, Munich, Germany. [13]EMBL-EBI, Wellcome Genome Campus, Hinxton, Cambridgeshire, UK. [14]Syngenta Crop Protection, Research Triangle Park, Durham, NC, USA. [15]Plant Science Program, Biological and Environmental Science and Engineering Division, King Abdullah University of Science and Technology (KAUST), Thuwal, Saudi Arabia. [16]Crop Plant Genetics, Institute of Agricultural and Nutritional Sciences, Martin Luther University of Halle-Wittenberg, Halle (Saale), Germany. [17]Leibniz Institute of Plant Genetics and Crop Plant Research (IPK) Gatersleben, Seeland, Germany. [18]School of Life Sciences, Technical University Munich, Freising, Germany. [19]Crop Development Centre, The University of Saskatchewan, Saskatoon, SK, Canada. [20]Centre for Crop & Food Innovation, Food Futures Institute, Murdoch University, Murdoch, WA, Australia. [21]School of Biological Sciences, University of East Anglia, Norwich, UK. [49]These authors contributed equally: Benjamen White, Thomas Lux, Rachel Rusholme-Pilcher, Ángéla Juhász. ✉e-mail: manuel.spannagl@helmholtz-muenchen.de; anthony.hall@earlham.ac.uk

## 10+ Wheat Genome Project

Sean Walkowiak[19,22], Valentyna Klymiuk[19], Brook Byrns[19], Kirby Nilsen[19], Jennifer Ens[19], Krystalee Wiebe[19], Amidou N'Diaye[19], Pierre J. Hucl[19], Curtis J. Pozniak[19], Bin Xiao Fu[22], Liangliang Gao[23], Emily Delorean[23], Dal-Hoe Koo[23], Allen K. Fritz[23], Jesse Poland[23], Cecile Monat[17], Axel Himmelbach[17], Anne Fiebig[17], Sudharsan Padmarasu[17], Uwe Scholz[17], Martin Mascher[17], Nils Stein [16,17], Georg Haberer[2], Heidrun Gundlach [2], Klaus F. X. Mayer [2,18], Manuel Spannagl[2,20]✉, Mulualem T. Kassa[24], Pierre Fobert[24], Sateesh Kagale[24], Jemima Brinton[4], Ricardo H. Ramirez-Gonzalez[4], Michael Bevan[4], Neil McKenzie[4], Burkhard Steuernagel[4], Cristobal Uauy [4], Markus C. Kolodziej[25], Simon G. Krattinger[15,25], Beat Keller[25], Thomas Wicker[25], Dinushika Thambugala[26], Curt A. McCartney[26], Venkat Bandi[27], Jorge Nunez Siri[27], Carl Gutwin[27], Catharine Aquino[28], Masaomi Hatakeyama[7,28], Dario Copetti[7,29], Gwyneth Halstead-Nussloch[7], Timothy Paape[7], Rie Shimizu-Inatsugi[7], Kentaro K. Shimizu[7,30], Tomohiro Ban[30], Kanako Kawaura[30], Toshiaki Tameshige[30], Hiroyuki Tsuji[30], Luca Venturini[31], Matthew Clark[31], Bernardo Clavijo[1], Christine Fosker[1], Gonzalo Garcia Accinelli[1], Darren Heavens[1], Ksenia Krasileva[1], David Swarbreck [1], Jonathan Wright [1], Anthony Hall [1,21]✉, Keith A. Gardner[32], Nick Fradgley[32], Lawrence Percival-Alwyn[32], James Cockram[32], Juan Gutierrez-Gonzalez[33], Gary Muehlbauer[33], Chu Shin Koh[34], Andrew G. Sharpe[34], Jasline Deek[35], Alejandro C. Costamagna[36], Hiroyuki Kanamori[37], Fuminori Kobayashi[37], Tsuyoshi Tanaka[37], Jianzhong Wu[37], Hirokazu Handa[10,37], Tony Kuo[38,39], Jun Sese[39,40], Kazuki Murata[41], Yusuke Nabeka[41], Shuhei Nasuda[41], Philomin Juliana[42], Ravi Singh[42], Hikmet Budak[43], Ian Small[44], Joanna Melonek[44], Sylvie Cloutier[45], Gabriel Keeble-Gagnère[46], Josquin Tibbets[46], Erik Legg[14], Arvind Bharti[14], Peter Langridge[47], Ken Chalmers[47] & Assaf Distelfeld[48]

[22]Grain Research Laboratory, Canadian Grain Commission, Winnipeg, MB, Canada. [23]Department of Plant Pathology, Kansas State University, Manhattan, KS, USA. [24]Aquatic and Crop Resource Development, National Research Council Canada, Saskatoon, SK, Canada. [25]Department of Plant and Microbial Biology, University of Zurich, Zurich, Switzerland. [26]Morden Research and Development Centre, Agriculture and Agri-Food Canada, Morden, MB, Canada. [27]Department of Computer Science, University of Saskatchewan, Saskatoon, SK, Canada. [28]Genomics/Transcriptomics group, Functional Genomics Center Zurich, Zurich, Switzerland. [29]Institute of Agricultural Sciences, ETHZ, Zurich, Switzerland. [30]Kihara Institute for Biological Research, Yokohama City University, Yokohama, Japan. [31]Life Sciences Department, Natural History Museum, London, UK. [32]The John Bingham Laboratory, NIAB, Cambridge, UK. [33]Department of Agronomy and Plant Genetics, University of Minnesota, Saint Paul, MN, USA. [34]Global Institute for Food Security, University of Saskatchewan, Saskatoon, SK, Canada. [35]School of Plant Sciences and Food Security, Tel Aviv University, Ramat Aviv, Israel. [36]Department of Entomology, University of Manitoba, Winnipeg, MB, Canada. [37]Institute of Crop Science, NARO, Tsukuba, Japan. [38]Centre for Biodiversity Genomics, University of Guelph, Guelph, ON, Canada. [39]National Institute of Advanced Industrial Science and Technology (AIST), Tokyo, Japan. [40]Humanome Lab, Tokyo, Japan. [41]Laboratory of Plant Genetics, Graduate School of Agriculture, Kyoto University, Kyoto, Japan. [42]Global Wheat Program, International Maize and Wheat Improvement Center (CIMMYT), Texcoco, Mexico. [43]Montana BioAg, Missoula, MT, USA. [44]Australian Research Council Centre of Excellence in Plant Energy Biology, School of Molecular Sciences, University of Western Australia, Perth, WA, Australia. [45]Ottawa Research and Development Centre, Agriculture and Agri-Food Canada, Ottawa, Ontario, Canada. [46]Agriculture Victoria, AgriBio, Centre for AgriBioscience, Bundoora, VIC, Australia. [47]School of Agriculture, Food and Wine, University of Adelaide, Adelaide, SA, Australia. [48]Institute of Evolution and Department of Evolutionary and Environmental Biology, University of Haifa, Haifa, Israel.

