## [Peer Review file · Nature Communications]

De Novo Annotation Reveals Transcriptomic Complexity Across the Hexaploid Wheat Pan-Genome

Corresponding Author: Professor Anthony Hall

Version 0:

Reviewer comments:

Reviewer #1

(Remarks to the Author)

The authors have annotated and compared multiple high quality wheat genome assemblies and examined specific gene classes. The approach permitted the analysis of tandem duplications, features that are known to have agronomic importance. While the manuscript provides a valuable resource, the authors make claims of generating a pangenome. I do not see evidence of this in the manuscript, they compared annotated genes but the approach they used does not produce a pangenome. There are pangenomes available for bread wheat that include some of the lines used here so it would be useful to compare gene presence absence variation between these studies to support their claims. The authors use the term pan cultivar – they are cultivars so no need for the term ‘pan’.

The manuscript seems unfinished with some edits tracked. The citations also do not seem related to the text, for example they cite 4 and 24 as previous wheat pangenome papers but neither paper describes a pangenome.

The manuscript has potential as a wheat resource, however they should perform comparison of their annotations with published wheat pangenomes and avoid claiming to have made a pangenome when one is not presented. I suggest that the authors undertake a more detailed review of related publications in this area and ensure their citations are related to the text.

Reviewer #3

(Remarks to the Author)

The authors have made significant improvements to the manuscript, and the detailed responses to the reviewers' comments are appreciated. However, there are still a few minor issues that need to be addressed to enhance clarity, consistency, and accuracy. Below are my specific comments and suggestions:

The authors only responded to the major comment (e.g., Q7 regarding to four accessions used for co-expression analysis) but did not revise the main manuscript accordingly. The “nine wheat cultivars” used for pangenome, while only four are used for co-expression analysis. This inconsistency may confuse readers unless a clear and simplified explanation is provided. Figure 1B: Norin 61 shows a notably larger proportion of “above_3” variants compared to other cultivars. Is this due to its unique breeding history, methodological bias, or other factors? Also, the meanings of “above_3” and “exact_dup” should be clarified in the figure legend.

Figure 2D: Please confirm whether the statistical test was performed correctly, especially since the p-value is shown as 0 for the expression comparison between core and shell genes in A, B, and D subgenomes.

GO analysis: Lines 198–215: The GO analysis was performed on only one accession to represent cultivar-specific genes. A single accession cannot represent cultivar-specific genes across all cultivars. What about using the remaining accessions individually for this analysis?

Line 94: *Brassica napus* should be italicized.

Figure 1B: The phenotype image is unclear.

Figure 2B: It is difficult to interpret due to missing legend information.

Line 126: Please clarify what defines a “high-confidence” gene module.

Lines 129–130: Abbreviations should be introduced at first mention, possibly around line 126.

Line 269: “t-test” – the “t” should be italicized.

Line 315: “de novo” should be italicized.

Line 335: Please clarify whether “OG” or “GO” is correct.

Line 484: Citation number should be superscript.

Lines 547–548: The manuscript still has some formatting issues after revision.

Version 1:

Reviewer comments:

Reviewer #1

(Remarks to the Author)

While the authors have addressed some of my comments, they seem unwilling to address others. The main issue is the claim of constructing a pangenome. The authors have compared gene content using orthofinder and claim this is a pangenome because other papers have been published with such erroneous claims. There is no pangenome constructed/available for download/reanalysis in this study so we should avoid perpetuating this error in terminology.

Similarly, the term pan transcriptome is confusing as such a term is very poorly defined and means different things to different people. The authors compare the expression of predicted core and variable genes which is fine but they should avoid using or misusing terminology.

(Remarks on code availability)

Reviewer #3

(Remarks to the Author)

All my previous concerns have been adequately addressed in the revised manuscript. I have no further comments.

(Remarks on code availability)

REVIEWER COMMENTS

We would like to thank the reviewers for their valuable comments and suggestions on our manuscript. Below, we provide detailed responses to all reviewers' individual queries.

Reviewer #1 (Remarks to the Author):

The authors have annotated and compared multiple high quality wheat genome assemblies and examined specific gene classes. The approach permitted the analysis of tandem duplications, features that are known to have agronomic importance.

While the manuscript provides a valuable resource, the authors make claims of generating a pangenome. I do not see evidence of this in the manuscript, they compared annotated genes but the approach they used does not produce a pangenome. There are pangenomes available for bread wheat that include some of the lines used here so it would be useful to compare gene presence absence variation between these studies to support their claims. The authors use the term pan cultivar – they are cultivars so no need for the term ‘pan’.

We apologize for being imprecise about our approach taken to define dispensable and conserved gene content between the reported wheat cultivars in our study. We now explicitly refer to a “gene-based pangenome”, with our approach in line with the definitions for a gene-based pangenome described and reviewed in Kaur et al. (*A stepwise guide for pangenome development in crop plants: an alfalfa (Medicago sativa) case study. BMC Genomics 25, 1022 (2024).* <https://doi.org/10.1186/s12864-024-10931-w>) and elsewhere.

We followed the suggestion of the reviewer and compared the gene presence/absence (core/shell/cloud gene portions) results of our study with the respective results reported by Montenegro et al. 2017 which include some of the cultivars also described in this study. Absolute proportions of core-, shell- and cloud- genes are similar between the two studies, as well as with the numbers reported by Jiao et al. (2024). We added this observation to the main text in the respective section. While we also find overlap in genes contained in the core and shell portions of both studies, in our view the very different gene prediction approaches used both complicate and hinder any insightful comparisons. Instead, we believe that both approaches yield complementary results, where our study contributes gene models constructed on the basis of native transcriptome data as supporting evidence.

We followed the reviewer’s suggestion and changed the term “pan cultivar” to “cultivar” throughout the manuscript.

The manuscript seems unfinished with some edits tracked. The citations also do not seem related to the text, for example they cite 4 and 24 as previous wheat pangenome papers but neither paper describes a pangenome.

The track changes were left in deliberately as part of the resubmission process. We have carefully checked all the references and they are all correct. We have revised the descriptions of datasets to provide more explicit definitions, rather than relying on the term

'pangenome'. This is an issue affecting the entire genomics community, with the term 'pangenome' being applied to a limited-sized data set.

Reviewer #3 (Remarks to the Author):

The authors have made significant improvements to the manuscript, and the detailed responses to the reviewers' comments are appreciated. However, there are still a few minor issues that need to be addressed to enhance clarity, consistency, and accuracy. Below are my specific comments and suggestions:

The authors only responded to the major comment (e.g., Q7 regarding to four accessions used for co-expression analysis) but did not revise the main manuscript accordingly. The “nine wheat cultivars” used for pangenome, while only four are used for co-expression analysis. This inconsistency may confuse readers unless a clear and simplified explanation is provided.

We have revised this section both in the main manuscript and the Methods section and clarified our reasons for selecting four of the cultivars for further co-expression analysis. We explain that these cultivars (ArinalrFor, Jagger, Julius and Norin 61) were selected as they encompass the range of ancestral groups represented by the wheat pan-genome modern cultivars (AG 1, 2 and 5) identified in doi.org/10.1038/s41586-024-07682-9.

Figure 1B: Norin 61 shows a notably larger proportion of "above_3" variants compared to other cultivars. Is this due to its unique breeding history, methodological bias, or other factors? Also, the meanings of “above_3” and “exact_dup” should be clarified in the figure legend.

Thank you for spotting this. We have re-computed BUSCO scores for both the genome assembly and gene annotation and confirm this observation in both for Norin 61. Therefore, a methodological bias or annotation issue is unlikely, unless already present in the initial sequence assembly. We believe the unique breeding history of Norin 61 with the reported introgressions (also described in this manuscript) could have contributed to this observation, but a more profound analysis of the underlying segments etc would require additional experimental/sequencing work. We clarified the meaning of the two categories in the figure legend.

Figure 2D: Please confirm whether the statistical test was performed correctly, especially since the p-value is shown as 0 for the expression comparison between core and shell genes in A, B, and D subgenomes.

We have amended this figure to show the correct adjusted p-values for these comparisons.

GO analysis: Lines 198–215: The GO analysis was performed on only one accession to represent cultivar-specific genes. A single accession cannot represent cultivar-specific genes across all cultivars. What about using the remaining accessions individually for this analysis?

The section describing Norin specific genes exemplifies the characteristics of putatively introgressed regions and provides a more detailed analysis of expression patterns, gene & TE content and functional descriptions. We also provide the results of the pangenome-wide GO enrichment of the core-, shell- and cloud compartments in Table S3 (in text: “In the set of cloud genes, functions related to chromatin organisation and reproductive processes were found to be enriched (**Table S3**)”).

We believe that the enrichment analysis results for the cultivar-specific genes of this single accession Norin 61 are useful and relevant in the context of the introgressed regions.

Line 94: *Brassica napus* should be italicized. Done

Figure 1B: The phenotype image is unclear.

We apologize for the poor resolution of these images in the current draft. We will provide higher resolution versions of these images for the production manuscript.

Figure 2B: It is difficult to interpret due to missing legend information.

We have added further detail to the legend for Figure 2B to aid interpretation.

Line 126: Please clarify what defines a “high-confidence” gene module.

We added a reference in the respective line to the Methods section where we specify the precise criteria for the confidence classification (high- and low- confidence) of our *de novo* gene predictions.

Lines 129–130: Abbreviations should be introduced at first mention, possibly around line 126. Done

Line 269: “t-test” – the “t” should be italicized. Done

Line 315: “de novo” should be italicized. Done

Line 335: Please clarify whether “OG” or “GO” is correct. The usage of OG is correct here.

Line 484: Citation number should be superscript. Done

Lines 547–548: The manuscript still has some formatting issues after revision. Amended